

# Survey data of damaged residential buildings and economic activities from the 2022 record-breaking flood in the Marche region, Italy

Sara Rrokaj[1], Chiara Arrighi[2], Marta Ballocci[1], Gabriele Bertoli[2], Francesca da Porto[3], Claudia De Lucia[2], Mario Di Bacco[2,5], Paola Di Fluri[4], Alessio Domeneghetti[4], Marco Donà[3], Alice Gallazzi[1], Andrea Gennaro[3], Gianluca Lelli[4], Sara Mozzon[3], Natasha Petruccelli[4], Elisa Saler[3], Anna Rita Scorzini[5], Simone Sterlacchini[6], Gaia Treglia[1], Debora Voltolina[6], Marco Zazzeri[6], Daniela Molinari[1]

[1]Department of Civil and Environmental Engineering, Polytechnic of Milan, Milan, 20133, Italy
[2] Department of Civil and Environmental Engineering, University of Florence, Florence, 50139, Italy
[3] Department of Geoscience, University of Padua, Padua, 35131, Italy
[4] Department of Civil, Chemical, Environmental and Material Engineering, Alma Mater Studiorum University of Bologna, Bologna, 40126, Italy
[5] Department of Civil – Environmental and Architectural Engineering, University of L'Aquila, L'Aquila, 67100, Italy
[6] Institute of Environmental Geology and Geoengineering of the National Research Council (CNR-IGAG), Milan, 20131, Italy

*Correspondence to*: Sara Rrokaj (sara.rrokaj@polimi.it)

**Abstract.** Accurate flood damage data are essential for developing reliable flood risk assessments and designing effective risk management strategies. However, empirical flood damage data remain limited, particularly at the object level, hindering the calibration and validation of predictive models. Existing datasets are often highly aggregated and lack the granularity required for detailed analysis. This paper presents two comprehensive, micro-scale datasets documenting flood damage to 256 buildings, comprising both residential buildings and economic activities, surveyed in the aftermath of the 2022 flood event in the Marche region of Italy. The georeferenced datasets include information on hazard characteristics, buildings' vulnerability features, physical damage description across structural and non-structural components, indirect damage, and implemented mitigation measures. In addition, original survey forms are provided to support future data collections in different contexts. Datasets and survey forms are available at the link: https://doi.org/10.5281/zenodo.15591850. The quality and richness of these datasets make them a valuable resource for improving flood risk modelling and supporting local stakeholders in identifying intervention priorities.

## 1 Introduction and case study

The exposure of people and assets to flood risk has been increasing in recent decades due to the rising frequency and intensity of flood events. This trend highlights the urgent need for effective flood mitigation strategies to minimize the negative consequences of floods. Risk assessment plays a central role in supporting these strategies, serving as a foundation for strategic intervention plans, facilitating resource prioritization and enhancing post-disaster recovery efforts. Moreover, the estimation of potential damage is essential to designing cost-effective mitigation strategies.





A significant challenge in advancing flood risk management is the scarcity of empirical flood damage data (Endendijk et al.,

2023; Kellermann et al., 2020; Merz et al., 2010). Moreover, much of the available data is aggregated (Merz et al., 2010;
Meyer et al., 2009) limiting the ability to capture critical information on the vulnerabilities and characteristics of damaged
assets. This limitation can be even more relevant for certain exposed assets, like business activities, whose heterogeneity
requires highly detailed datasets that are often unavailable (Endendijk et al., 2024; Schoppa et al., 2020; Seifert et al., 2010;
Sultana et al., 2018).

The lack of detailed data not only hampers the development of robust flood damage models but also limits the validation of
existing ones, as the uncertainties associated with current methodologies are often insufficiently addressed (Apel et al., 2009;
Bubeck et al., 2011; Merz et al., 2010b; Molinari et al., 2019). In the absence of site-specific damage data and models, it is
common practice to use damage models developed for other geographical regions (Cammerer et al., 2013), further amplifying
the intrinsic uncertainties associated with damage estimation (Scorzini & Frank, 2017). In fact, flood damage models should

be tailored to the context of implementation, as damage features can vary significantly depending on local conditions.

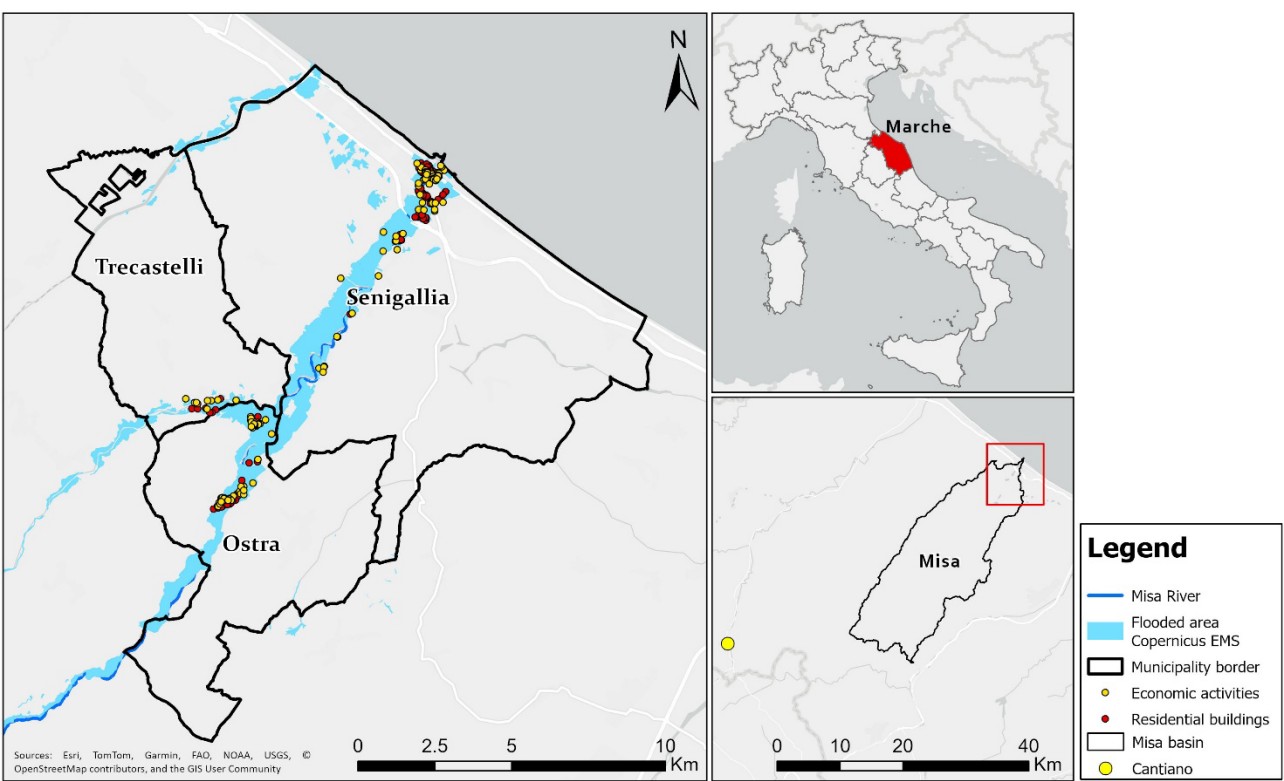

**Figure 1. Study area. Points in the left map depict the buildings surveyed. The flooded area has been provided by Copernicus Emergency Management Service (EMS).**

In Italy, the absence of a standardized damage data collection procedure (Molinari et al., 2014) and the lack of a national

damage database result in scarce and often incomplete information on flood damage to the residential and economic sectors
(Ballocci et al., 2024; Amadio et al., 2019; Scorzini & Frank, 2017; Minucci et al., 2016).



This study aims to address this gap by providing empirical data on the damage and vulnerability of both economic activities and residential buildings affected by the record-breaking flood event occurred in the Marche Region (Central Italy) on September 15, 2022. During this event, cumulative rainfall exceeded twice the historical maxima, causing widespread and
severe impacts across the region. The highest recorded precipitation occurred in the municipality of Cantiano, where 419 mm of rainfall fell within nine hours (for more information on the event see (Lucia et al., 2024; Santangelo et al., 2023)). The flood caused extensive damage to buildings and infrastructure and resulted in 13 fatalities. The damage data presented in this paper were collected through a field survey conducted in the municipalities of Ostra, Senigallia, and Trecastelli, located within the Misa River Basin (see Fig 1), between October and December 2022. This area features a varied socio-economic and
geographical landscape: the upstream zone, and particularly the municipality of Ostra, is primarily industrial, while the downstream area, including the city centre of Senigallia, is predominantly commercial.

The dataset presented in this study contributes to the development of more accurate flood damage models and supports local authorities in identifying intervention priorities across the territory. Furthermore, the data can assist practitioners in validating and refining existing damage models for the economic and residential sectors.

## 65 2 Methods

Data on damaged economic activities and residential buildings were collected using ad hoc forms - Forms A, B, C, D, and E - originally developed by Molinari et al. (2014) and tested in the Umbria region in 2012 and 2013. These forms follow the structure described in detail in Berni et al., (2017), which is summarized below for the two asset categories considered in this study.

Economic activities:

- Form A collects general information on the flood event, the physical vulnerability of the building, its location, the type of damages incurred by the activity, and the damage to employees.
- Form B collects detailed damage data of the building structure and systems, including floor-specific damage information.
- Form C gathers information on damages to machinery, production plants, equipment, furniture, storage areas, archives, and vehicles.
- Form D collects information on the recovery costs (e.g., clean-up costs) and mitigation actions undertaken before, during and after the event.
- Form E collects information on indirect damages and reimbursement mechanisms.

Residential buildings:

- Form A collects general information on the flood event, the physical vulnerability of the building, its location and the type of damages incurred, identifying affected parts of the building.





- Form B collects detailed data on damage and vulnerability of housing units, including floor-specific information. For each housing unit, undertaken mitigation measures and recovery costs are collected as well.

- Form C collects general information on damages to common areas, including flood-specific damage information.

- Form D collects the same information as Forms A and B, but for attached buildings.

- Form E collects information on structural damages, considering the building construction materials. Specifically, the From E is divided into two sub-forms: Form E1, which targets concrete and steel buildings, and Form E2 which is tailored for masonry and wooden buildings.

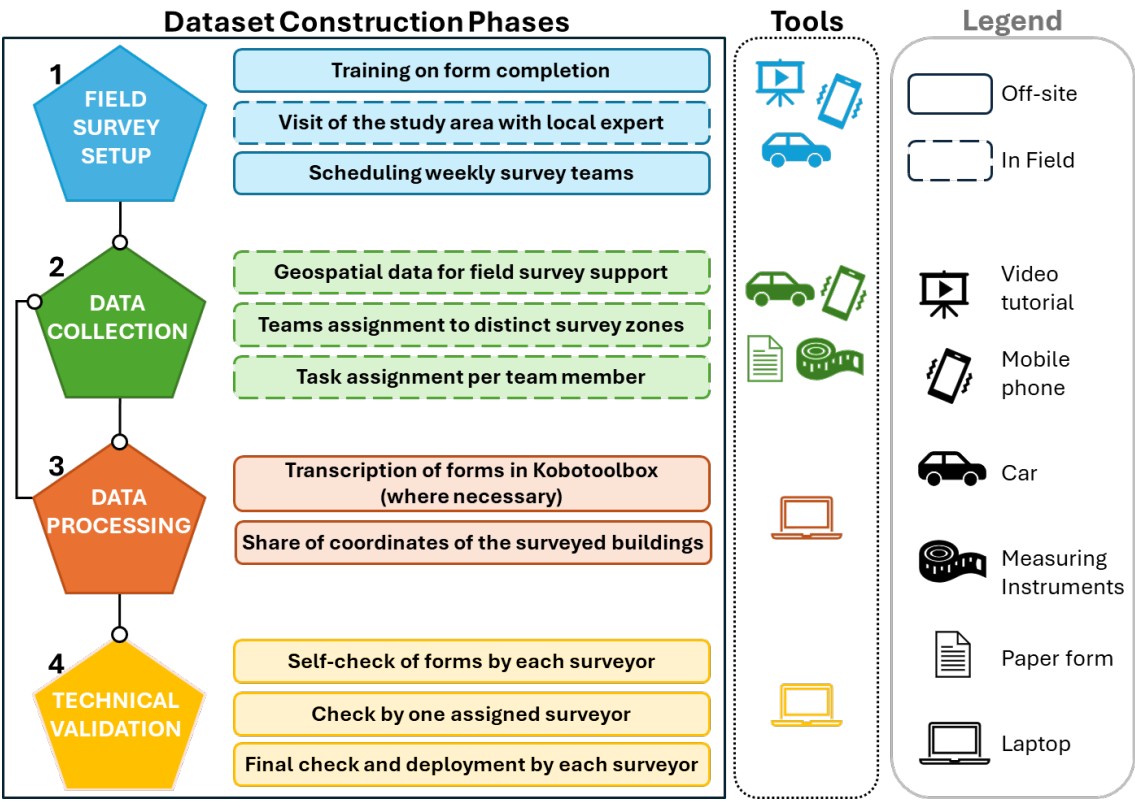


**Figure 2. Workflow of datasets construction and schematic representation of the tool used. Each pentagon represents a distinct phase in the data construction process, numbered according to the chronological order in which the phases were carried out. A legend for the graphical elements is provided on the right.**


The dataset creation process follows the workflow illustrated in Fig. 2. About 40 researchers from five Italian universities, University of L'Aquila, Politecnico di Milano, University of Bologna, University of Florence, and University of Padua, participated in the process. The process was structured into four main phases: field survey setup (Phase 1), data collection (Phase 2), data processing (Phase 3), and technical validation (Phase 4). Each phase was categorized as either field-based (in



situ) or office-based (off-site), as illustrated in the framework diagram. All phases were supported by specific equipment and tools. A detailed description of Phases 1 to 4 and the equipment used in each is provided below.

## 2.1 Phase 1 – field survey setup

Before the fieldwork, all participating researchers received training on form completion. Specifically, a video tutorial explaining the form compilation process was recorded and distributed to ensure methodological consistency. Moreover, a
reconnaissance field visit was conducted on October 10, 2022, guided by local experts to gain insights into the affected municipalities. The visit included meetings with the mayors of the three surveyed municipalities, on foot surveys in the city centre of Senigallia and a drive-through assessment of the upstream area, with targeted stops at locations that had suffered severe damage from the event. This preliminary activity helped secure the commitment and support of the local authorities and enable the identification of priority areas for data collection during the main field campaign. To organize the fieldwork
campaign, university representatives provided a list of participating researchers. The scheduling process was coordinated remotely, ensuring that at least two teams, each consisting of two or three members, were deployed weekly in the field. Decisions regarding which municipality to survey were made on a weekly basis to ensure balanced spatial coverage of affected assets and efficient logistical planning.

## 2.2 Phase 2 – data collection

The data collection phase was conducted in the immediate aftermath of the event, from October to December 2022. Survey teams in the field relied on geospatial data support, by integrating the flooded area map and the locations of surveyed building into Google Earth application on their smartphones. The flooded area map, provided by the Copernicus Emergency Management Service mapping component (available at https://mapping.emergency.copernicus.eu/activations/EMSR634/), enabled efficient navigation and focused data collection within the potentially affected zones. The inclusion (with continuous
updates) of previously surveyed economic activities and residential buildings prevented duplication of efforts by avoiding repeated surveys of the same structures.  Additionally, field teams divided their work into distinct geographic zones to further avoid overlap. Data collection was guided by specific forms developed by Molinari et al., (2014) (included in the public repository of this paper). These forms were made available to surveyors both in papery and digital version, leaving them free to choose the tool they found easier to use in the field, depending on the local circumstances. The digital version of the form
was developed by the Institute of Environmental Geology and Geoengineering of the National Research Council (CNR-IGAG) of Milan, Italy, in KoboToolbox (https://www.kobotoolbox.org/), an open-source platform for data management, which enabled efficient storage, organization, and preliminary analysis of the data. Within each team, members had defined roles, including taking pictures with the smartphone, conducting measurements, and filling in the survey forms. Each team completed the survey in 30-60 minutes depending on the size and complexity of the building.



## 2.3 Phase 3 – data processing

The data processing phase focused on digitizing (where necessary) and structuring the collected information. All the information gathered during the fieldwork in papery form were transcribed in the digital version by means of KoboToolbox, to create a complete, georeferenced and structured digital database. Moreover, surveyors recorded the coordinates of the surveyed buildings in a shared file. This file was continuously updated and used during the survey campaign (Phase 2) to ensure that the same building was not surveyed multiple times.

## 2.4 Phase 4 – technical validation

A technical validation process was implemented, for each surveyed building/activity, to ensure the reliability and consistency of the collected data. This process involved a three-step control procedure. First, for data collected in the field by means of papery forms, each surveyor reviewed their own notes to verify that all information were accurately transcribed into the digital forms in KoboToolbox. Second, an independent review was conducted, i.e. by a surveyor external to the field team, focusing on data coherence. This included checking for physically unfeasible values which could result from data entry errors. The review also identified duplicate entities by comparing location coordinates (e.g., multiple housing units in the same residential buildings entered in two separates forms). Finally, based on the outcomes of this review, surveyors were requested to correct or complete any missing, incorrect, or misleading information prior to submitting the finalized forms in KoboToolbox.

## 3 Data records

The two datasets – one on economic activities and the other on residential buildings – contain information on a total of 256 surveyed buildings. Being elaborated from the Kobotoolbox repository, each record in the datasets includes geographic coordinates, enabling a straightforward integration into a Geographic Information System (GIS) environment.

The structure of the economic activity and residential building datasets, along with their supporting documentation, is illustrated in Fig. 3 and 4, respectively. Each dataset consists of five Excel workbooks (Forms A to E), corresponding to distinct survey forms and named accordingly (e.g., Form_A.xlsx). Some of these workbooks are further subdivided in worksheets. Fig. 3 and 4 show the hierarchical organization of information within the workbooks, the interconnections among the different workbooks, and the relationship between workbooks and their corresponding worksheets. Both datasets follow a uniform hierarchical structure comprising three levels: section, aspect, and data. Sections aggregate variables under broad thematic domains. Within each section, aspects represent more specific subtopics. Within aspects, the data level contains the individual variables. This structure reflects the organization of the original survey forms. Each record in the workbooks is identified by a unique alphanumeric ID, consisting of the letter corresponding to the form and a sequential number (e.g., A1, A2, etc..). The ID of Form A workbook is also reported in Forms B, C, D, and E workbooks, allowing all records to be linked back to the general information collected in Form A.

The following subsections provide an overview of the dataset structure, describe the relationships among workbooks and worksheets, and outline the supporting documentation provided for the two datasets.

## 3.1 Economic activities dataset

The dataset on economic activities comprises 133 records, including production and commercial activities, and it is organized according to the structure depicted in Fig. 3. As shown, the Excel workbooks corresponding to Forms B, C, D, and E are linked

to Form A through a one-to-one relationship (1:1). This implies that each record in Form A may be associated with at most one record in each of Forms B, C, D, and E. Moreover, the structure shows that the data collected through the survey forms A, D and E are organized in a unique worksheet. Conversely, Forms B and C are both organized in two worksheets within their respective Excel workbooks. Regarding Form B, the first worksheet contains building-level information, while the second worksheet details floor-specific data. To connect the two, the ID of Form B reported in the first worksheet is included in the

second worksheet. Each record of the first worksheet corresponds to one or more floors (1:N relationship) detailed in the second worksheet. Regarding Form C, the first worksheet contains information on damages occurred to assets inside or outside the buildings (e.g., machinery or company vehicles), while the second worksheet provides detailed information on the characteristics of each damaged vehicle, with one record per damaged vehicle. Thus, if damage to company vehicles occurred, one to multiple records (1:N relationship) in the second worksheet are linked to a single record in the first worksheet via the

Form C ID.

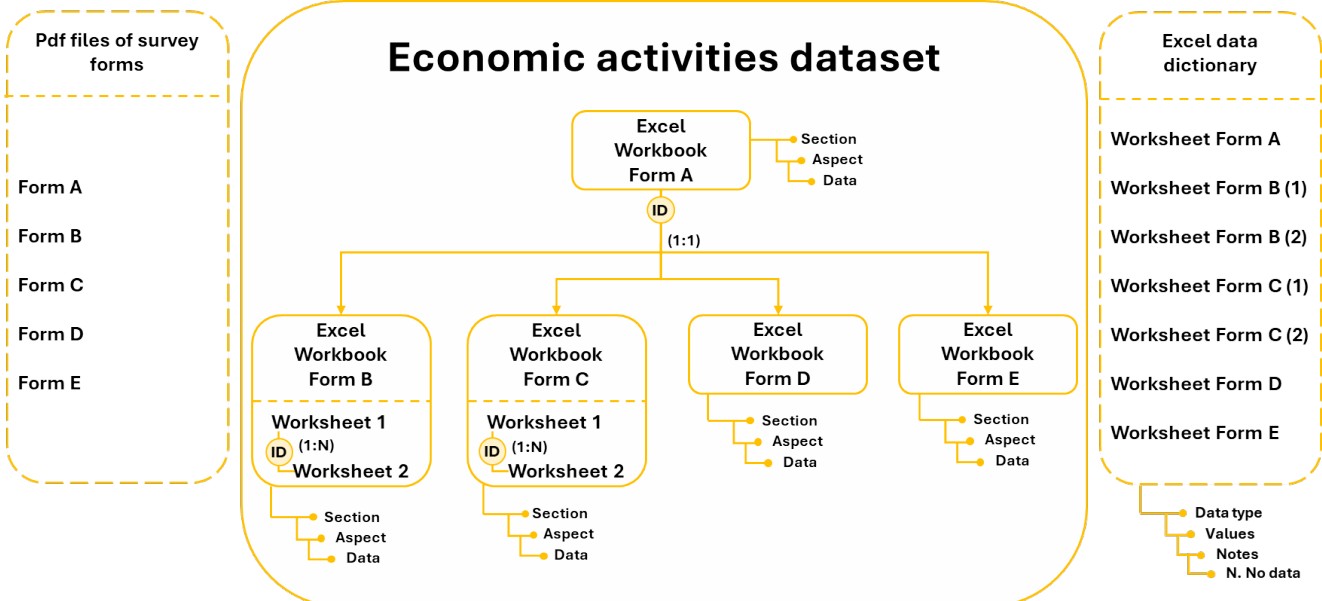

**Figure 3. Structure of the data provided for the economic activities. Solid lines depict the dataset; dashed lines indicate the supporting documentation. Excel workbooks are further subdivided into worksheets (central and right panels); if a workbook contains only a single worksheet, no further subdivision is shown. Survey forms are organized into separate Pdf files (left panel). The relationship**

**between Excel workbooks and their corresponding worksheets is indicated in parentheses.**



Consequently, the number of records in each workbook and its corresponding worksheets could differ depending on the type of damage occurred as well as the vulnerability characteristics of the activity. Workbook A contains 133 records; Workbook B contains 128 records in Worksheet 1 and 139 records in Worksheet 2; Workbook C contains 131 records in Worksheet 1 and 79 records in Worksheet 2; Workbook D and E contain 133 records, each.

### 3.2 Residential building dataset

The dataset on residential building comprises 123 surveyed buildings. As illustrated in Fig. 4, the Excel workbooks corresponding to Forms B, C, D, and E are linked to Form A through different types of relationships that reflect the nature of the information collected in each form. Form B gathers data on individual housing units, while Form D contains information on attached buildings. Since multiple housing units and attached structures can be associated with a single residential building, these forms are related to Form A through a one-to-many relationship (1:N). Conversely, Form C collects damage information occurred to common areas, and Form E details the structural damages affecting the main building. Given that this information is unique for each residential building, a one-to-one relationship (1:1) with Form A exists. Thus, each record in Forms C and E corresponds to one record in Form A.

Moreover, each workbook, except the one of Form A, is organized into two worksheets. The first worksheet of Form B workbook contains general information on individual housing units, while the second worksheet provides floor-specific data. Accordingly, a one-to-many (1:N) relationship exists between Worksheet 1 and Worksheet 2, as multiple floors can belong to a single housing unit.

For Form C workbook, the first worksheet reports the number of floors affected by flooding in common areas, whereas the second worksheet offers detailed information on the damage observed at each floor. This structure implies a (1:N) relationship between the two worksheets.

In Form D workbook, the first worksheet records vulnerability characteristics, damage, and event-specific information for attached buildings. The second worksheet details the damage at the floor level. A (1:N) relationship consistently applies here as well, since each attached buildings can have multiple floors affected.

Finally, the Excel workbook for Form E contains two independent worksheets that are not linked to one another. Worksheet 1 corresponds to Form E1 and collects damage data to reinforced concrete and steel buildings, while Worksheet 2 corresponds to Form E2 and gathers damage data to masonry and wooden buildings.

As for economic activities, the number of records in each workbook and worksheet may differ depending on the type of damage occurred and the vulnerability characteristics of the buildings or housing units. Workbook A contains 123 records; Workbook B contains 117 records in Worksheet 1 and 137 records in Worksheet 2; Workbook C contains 15 records in both Worksheet 1 and in Worksheet 2; Workbook D contains 45 records in Worksheet 1 and 41 in Worksheet 2; Workbook E contains one record in Worksheet 1 and 6 in Worksheet 2.



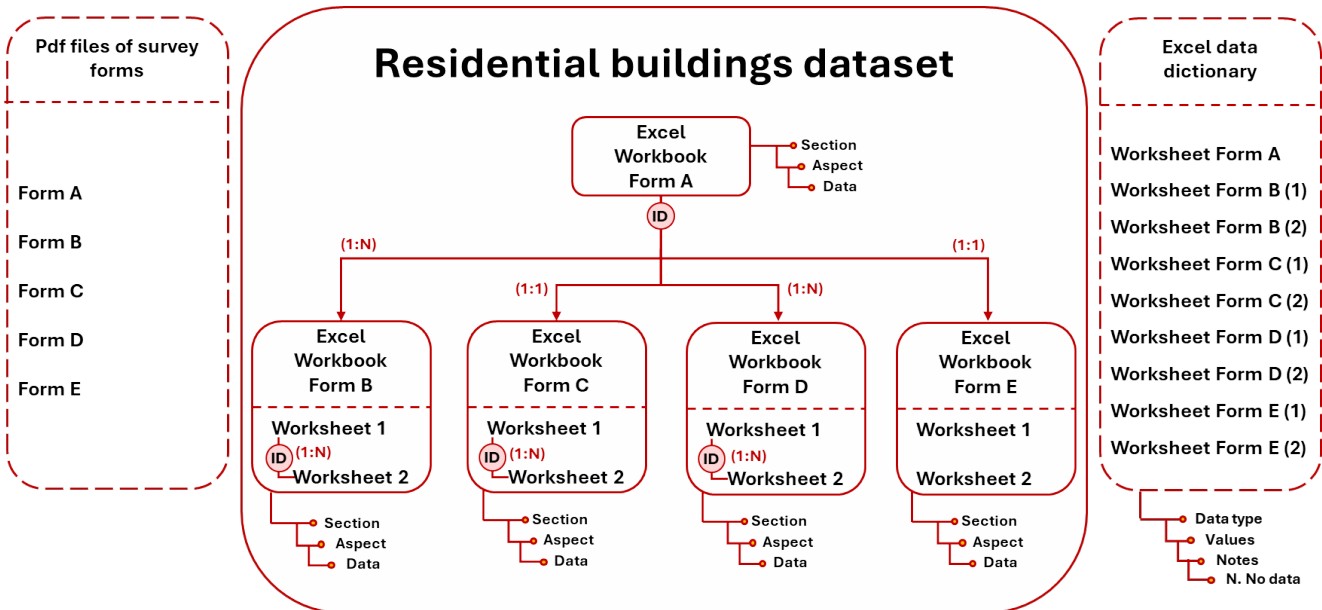

**Figure 4. Structure of the data provided for the residential buildings. Solid lines depict the dataset; dashed lines indicate the supporting documentation. Excel workbooks are further subdivided into worksheets (central and right panels); if a workbook contains only a single worksheet, no further subdivision is shown. Survey forms are organized in separate Pdf files (left panel). The relationship between Excel workbooks and their corresponding worksheets is indicated in parentheses.**

### 3.3 Supporting documentation

Both datasets are provided with supporting materials, including the original survey forms and a data dictionary. The survey forms are provided in 5 separate Pdf documents, corresponding to Forms A to E, as shown in Fig. 3 and Fig. 4.

The data dictionary is provided as an Excel workbook and summarizes the metadata associated with each variable at the data level. The workbook is organized into multiple worksheets corresponding to the individual data sheets in each dataset: 7 for the economic activities' dataset and 9 for the residential buildings' dataset. The data dictionary tables offer a description of each variable, detailing its name, data type, possible values with associated units of measurement, presence of notes, and the number of missing values (No data) of each data field. The information contained in the data dictionary are reported in Table 1.

**Table 1. Structure of data dictionary**

| Data dictionary fields | Description |
| --- | --- |
| Section | Name of the section reported in the worksheet |
| Aspect | Name of the aspect reported in the worksheet |
| Data | Name of the data reported in the worksheet |
| Type | Data type of each variable (float, integer, boolean, string, string binary or categorical) |





| Value | Data description and possible values or categories of each variable, including units of measurement where applicable |
|---|---|
| Notes | Indicates whether explanatory notes are provided for the aspect. "Yes" means at least one note is provided; "no" indicates none are available |
| No Data | Number of missing or unreported records for that variable |

It is important to clarify that missing data often result from conditional follow-up questions. Therefore, the absence of data does not necessarily indicate a missing response but may reflect that the follow-up question was not applicable and thus left intentionally unanswered.

## 4. Data availability

The data are available on the Zenodo platform at the following link: https://doi.org/10.5281/zenodo.15591850 (Rrokaj et al.,
235  2025).

## 5. Discussion

### 5.1 Usability and benefits of datasets

Through this manuscript we provide detailed structured damage data on economic activities and residential buildings, alongside the survey forms used to construct the datasets. Given the lack of existing context-specific and high-quality data,
these datasets offer several key benefits for a variety of uses.

For instance, at the event scale, these post-event damage datasets can help identifying territorial vulnerabilities and informing relevant stakeholders about the specific factors contributing to flood losses within business and residential sectors, to identify mitigation actions. At the building scale, the dataset links event characteristics and observed damage with detailed hazard and asset features, enabling a comprehensive understanding of the processes and vulnerability drivers that lead to damage; this
understanding is further facilitated by survey notes embedded in the datasets.

Fig. 5 provides an example of damage recorded at the building level. In detail, Fig. 5(a) shows the frequency of damages recorded for economic activities, disaggregated into commercial and industrial sectors, collected through Forms A and C, while Fig. 5(b) depicts the frequency of damages observed in residential housing units for each affected floor, collected through Form B. For economic activities, the plots clearly show that structural damage was reported less frequently, while damage to
goods in stocks as well as indirect impacts (such as inaccessibility and disruption of operations) were more commonly observed. Regarding residential buildings, a wide range of damage types was frequently reported, with the most common being direct damage to furniture, electrical appliances, doors and windows, technological systems, as well as indirect damage related to clean-up cost.

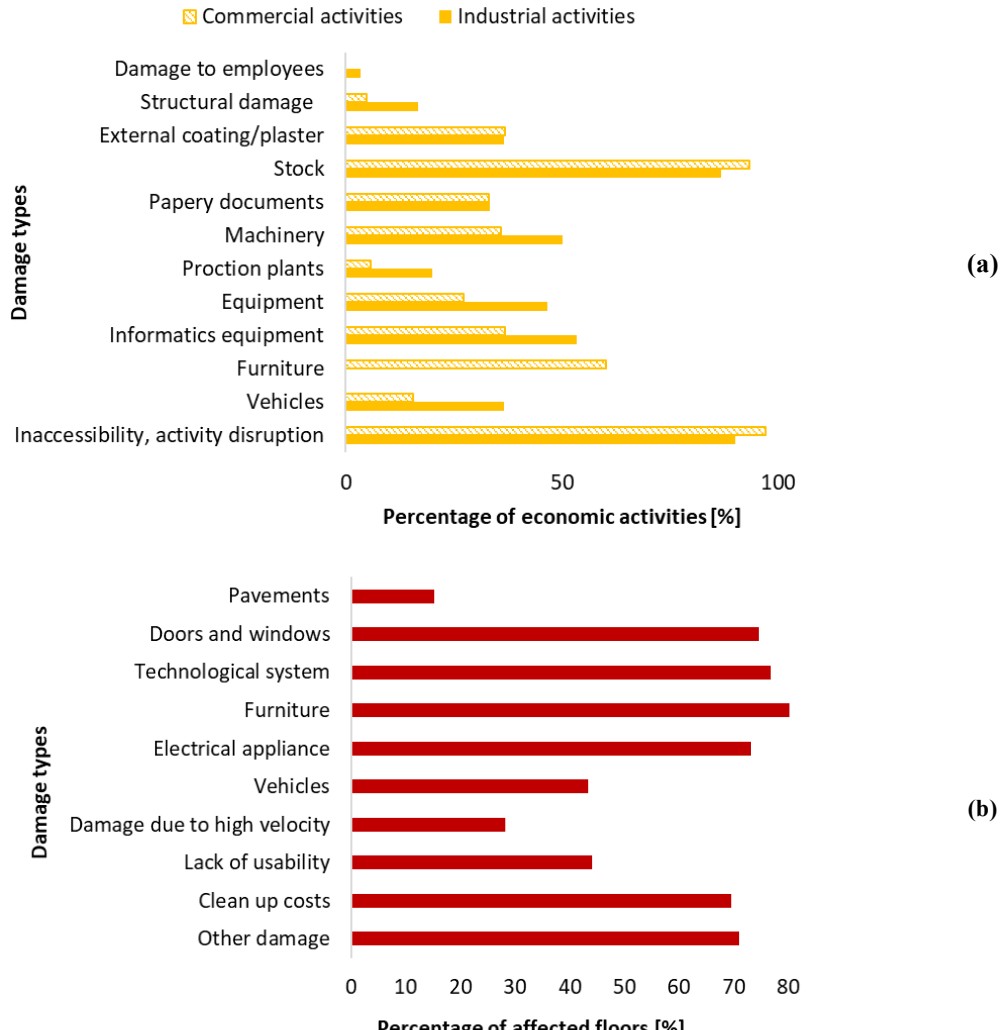

**Figure 5. (a) Types of damage collected for economic activities, disaggregated by commercial (n = 103) and industrial (n = 30) sectors, based on data from Workbooks A and C. (b) Damage types collected for residential buildings floors (n = 137), based on data from Worksheet 1 of Workbook B.**

The level of detail of this information could support the design of targeted mitigation strategies and communication campaigns
aimed at raising public awareness about common types of damage. In this context, learning from past events can encourage the adoption of preventive measures.

Moreover, these datasets can serve for developing and calibrating context-specific models. Specifically, the high level of detail of damage, hazard and vulnerability information collected at the building scale can provide a solid foundation for developing





advanced micro-scale damage models for the Misa basin area. In addition, the survey data, when combined with ancillary
information such as damage compensation claims, can also be used to validate existing damage models developed in other
regions.

With this regard, it is worth noting that the dataset on economic activities has been already used by (Ballocci et al., 2024) for
direct damage estimation. Nevertheless, it is important to mention that beyond direct impacts the datasets include information
on indirect damage (e.g., clean-up costs, the duration of business interruption and periods when buildings were unusable),
which are often overlooked in existing assessments but are essential for the development of more comprehensive damage
models.

The datasets also include specific information on hazard features (e.g., maximum water depth, flood water permanence inside
the buildings, presence and type of sediments and contaminants) at 256 spatial locations within the flooded area (see Fig. 6).
This detailed data can be used to reconstruct the flood event accurately and validate the results of hydrodynamic simulations.
Such data are particularly valuable in the case of the Misa event, where the flood wave damaged monitoring instruments,
preventing the collection of official measurements.

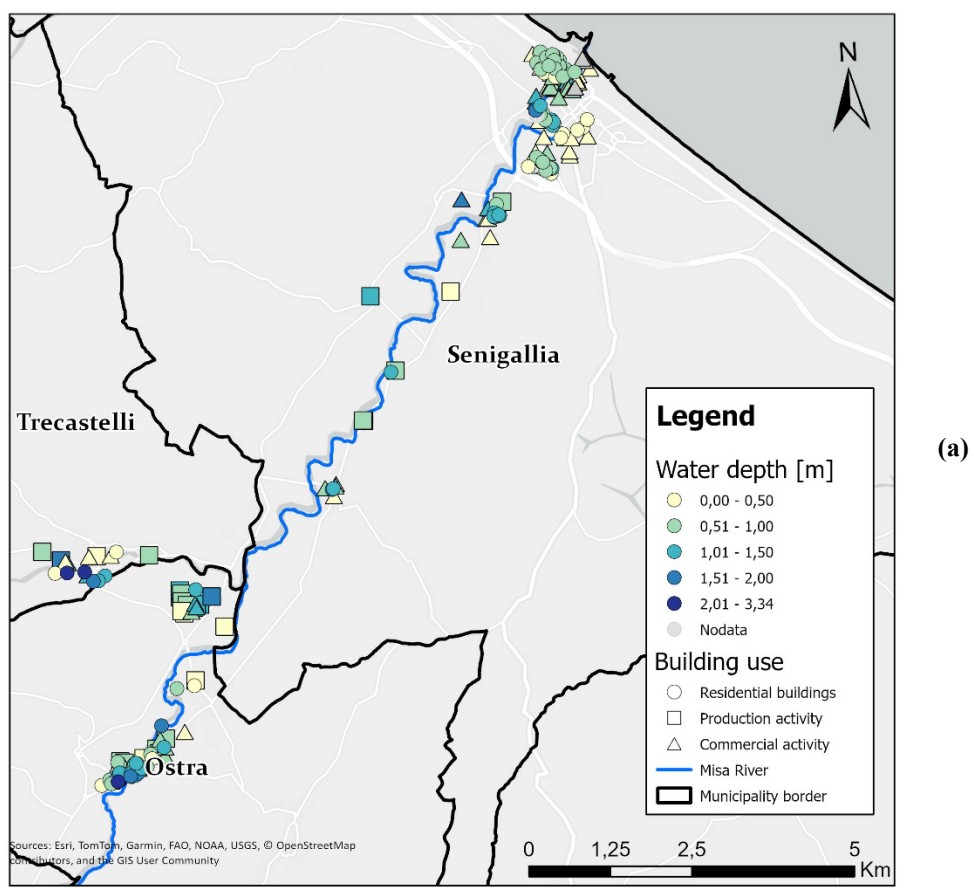





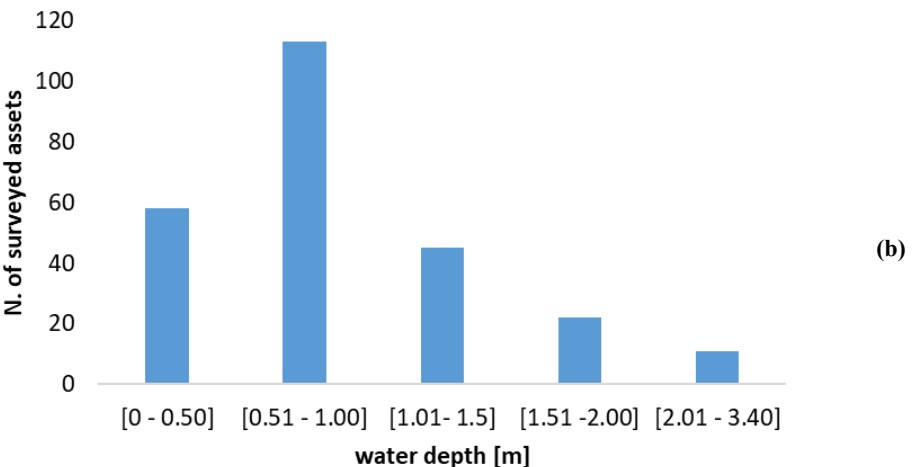

**Figure 6. (a) Map of water depth distribution across the study area disaggregated by building type: residential, commercial, industrial. (b) Water depth frequencies for the two datasets. Water depth was available for 247 out of 256 buildings.**


Finally, the publication of the survey forms not only supports the development of complete and high-quality datasets in other contexts but also facilitates their adaptation to different hazards and asset types. Their versatility across diverse scenarios is demonstrated by their history of use: originally developed and applied in the Umbria region in 2013, the forms were later employed to collect damage data in the Marche region (as presented in this manuscript), and subsequently adapted for a survey

conducted among flooded households in Belgium and the Netherlands (Rodríguez Castro et al., 2025)

**5.2 Datasets limitation**

Despite the high quality of the datasets, two main limitations should be highlighted. First, some variables could not be recorded either because the information was difficult to retrieve (e.g., year of construction) or not reported during the field survey (e.g., for human error). Second, a significant portion of the affected assets was not included in the survey. Although efforts were

made to ensure significant coverage in terms of representativeness of the surveyed assets in terms of hazard, exposure and vulnerability characteristics, the complete assessment of all damaged assets would have required a considerable amount of time and resources. Survey coverage was also affected by the accessibility of properties during the data collection period. In fact, in the most severely impacted areas, some residences and businesses remained unoccupied and visibly damaged even nearly three months after the event. This was likely influenced by the timing of the flood, which occurred in mid-September;

the arrival of colder weather soon after the event hampered clean-up and recovery efforts, leading many residents to delay restoring their homes until the following spring. The same applies to some commercial activities, with several business owners indicating their intention to postpone recovery until the start of the tourist season.



**Author contribution**

SR: data collection, data curation, original draft preparation; CA: data collection, review and editing; MB: data collection; GB: data collection; FP: reviewing and editing; CDL: data collection; MDB: data collection; PDF: data collection; AD: data collection, review and editing; MD: data collection; AG: data collection; AGe: data collection; GL: data collection; SM: data collection; NP: data collection; ES: data collection; ARS: data collection, review and editing; SS: data collection, review and editing; GT: data curation; DV: software; MZ: software; DM: data collection, review and editing and supervision.

**Competing interests**

The authors declare that they have no conflict of interest.

**Acknowledgements**

The authors gratefully acknowledge:

- Prof. M. Brocchini (professor of Polytechnic University of Marche) for the support in organizing data collection and
the guide during the reconnaissance field visit;
- Federica Fanesi (mayor of Ostra), Massimo Olivetti (mayor of Senigallia), and Marco Sebastianelli (mayor of Trecastelli) for their support during the field survey;
- Mattia Magni, Mattis Sormanni, Leonardo Valagussa (students of Polytechnic of Milan) for their help in data collection;
- Barbara Polidoro and ARUP team for their help in data collection;
- Simona Bernasconi for her help in data collection.

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
