# Peer review of "Survey data of damaged residential buildings and economic activities from the 2022 record-breaking flood in the Marche region, Italy"

_Earth System Science Data, 2025_

## Author Comment (AC1)

**Response to Reviewer #1**

We thank the reviewer for the constructive comments and feedback. Below, we provide point-by-point responses to each comment.

Original review comments are in **black,** and our responses are in **blue.** *Changes to be introduced in the revised version of the manuscript are shown in italics.*

**General comment:**

This is the first time that such detailed, micro-scale flood damage data has been made publicly available. The dataset is accessible through the provided links and is clearly described and well-organized in the attached Excel sheets, with separate documentation for both commercial and residential buildings. In its current form, the data can be used in a variety of contexts, including cross-validation of flood damage models, both residential and commercial, for other regions and countries, improving existing models, and identifying overlooked damage mechanisms.

It is noteworthy that several Italian universities have collaborated to develop a common survey methodology and a standardized dataset for post-event flood damage data collection. I hope that the aim is to maintain this effort over time, thereby creating a longitudinal dataset that supports the continuous improvement and adaptation of damage models to reflect the evolving physical and economic vulnerabilities of exposed assets, as well as to enable their validation.

**Specific comments:**

- **Section 2.2: Data collection**

Data collection is reported to have started immediately after the event. In contrast, previous studies that relied on post-event data collection typically began 6–8 months later, or even beyond that. This delay was intentional, allowing people time to reconstruct their buildings so that when surveys were conducted, most or all buildings would have been fully reconstructed, enabling a more accurate assessment of the original damage.

This difference in timing should be considered a limitation of the current dataset. Since data were collected immediately, the reported damage may not capture the full cost of damages that become apparent only during or after reconstruction efforts.

We thank the reviewer for raising this point about the timing of the data collection campaign. We agree that collecting data immediately after the event limited the possibility to capture certain types of damages, especially indirect ones, and led to an underrepresentation of damages to inaccessible and closed buildings.

In Section 5.2 (Data limitations) of the manuscript, the partial coverage of all damaged assets is already acknowledged; however, we will further expand this section with the

consideration on the potential underestimation of the indirect damages (e.g., damage caused by the humidity) for the surveyed assets, due to the timing of the survey.

Nevertheless, it is worth mentioning that the immediate post-event survey was essential to record accurate information that would have been difficult to obtain later. For instance, the visible flood marks at the time of the survey enabled precise water depth measurements. Moreover, collecting testimonies shortly after the event ensured that direct damage reports were not affected by memory omissions, since damages were clearly observable in the field at that time.

The following edits and the new sentence will be incorporated in the revised version of the manuscript:

> Line 287: *Despite the high quality of the datasets, **three** main limitations should be highlighted.*

> Line 289: *Second, the timing of the survey may have resulted in a potential underestimation of indirect damages, such as those caused to furniture, coating and plasters by humidity.*

- **Section 2.4: Technical validation**

It is very good that the collected data is also reviewed by an external team, in addition to the original data collection team. However, it is still unclear whether this review is conducted only on the paper forms before they are entered into KoboToolbox, or if it also includes forms that have already been digitized in the platform. If the review is limited to the paper forms, it is important to also double-check the digital entries. In previous studies, reviewing the digitized data has proven useful in identifying additional typing errors.

> The validation process was conducted in three steps. The first step involved only the data collected using paper forms, as during this phase the teams checked the data before entering it into KoboToolbox. The second step was carried out by a single reviewer, who performed a quality check directly in KoboToolbox after all data had been digitized on the platform. Thus, this second step was conducted on the entire dataset. The third step took place after the second review, during which all teams were asked to correct and complete missing information in KoboToolbox. Thus, only the first step was limited to the data originally collected on paper forms.

> In the revised manuscript, we will clarify that the complete datasets were involved in the second validation step, as follows:

> > Lines 140-141: *Second, an independent review was conducted, i.e. by a surveyor external to the field team, focusing on data coherence after all the paper forms were digitalized in KoboToolbox (Phase 3).*

**Minor comments:**

**1. Introduction and case study**

- Figure 1. Avoid using a yellow dot to represent the municipality of Catarino, as it is very similar to the one representing the economic activities. Please change for another colour.

- Figure 1. We are missing the representation of the three surveyed municipalities within the Misa River basin. Would it be possible to include them in the figure showing the basin? This would help illustrate which part of the basin was surveyed.

- Figure 1. In the legend, specify as in the caption that the economic and residential buildings in the municipalities are the surveyed ones.

  We thank the Reviewer for the suggestions; we will updated Figure 1 accordingly:

[Figure]

*Figure 1. (a) Marche region shown in red; (b) location of the three municipalities of Ostra, Senigallia, and Trecastelli surveyed within the Misa Basin, and the municipality of Cantiano, indicated with a green dot; (c) points representing the buildings surveyed within the three municipalities, and the flooded area of the 2022 event, as provided by the Copernicus Emergency Management Service (EMS).*

**2. Methods**

In the excel data dictionary:

Economic activities

Form A:

- It is not very clear to me the differences among the 4 variables representing building elevation (ΔQ, hg, h1 and h2).

  In the PDF of FORM A, there is a picture illustrating what these variables refer to. Specifically, the figure shows a reference survey point, indicated by a black horizontal dashed line, from which all measurements are taken. This is also the point where the external water depth is measured. To clarify the four measurements and support other researchers and practitioners in collecting these values in the field, we will provide an additional measurement guide uploaded with the datasets in Zenodo. This guide is attached to this response (see page 6).

- It would be valuable for future work to include an additional sediment variable representing large objects (e.g., tanks, cars, rubble from other buildings), as these objects could cause additional damage to building structures upon impact.

  We thank the Reviewer you for the suggestion. This aspect was included in the survey forms under the field *"other"*, where surveyors could report the presence of large objects.

Residential buildings

- In the dictionary of the database, it is not very clear the distinction of B, C and D forms, specify there too that B is for the housing unit, C is for the common areas and D for attached buildings.

  We will add a sentence to each worksheet of the data dictionary to clarify the distinction between the forms.

Form B, C, D:

- This form in the floor section includes a variable 'damage due to high velocity' how is this collected? Based on people perception? How do you double check this assumption?

  Floodwater velocity was assessed based on the interpretation of people's descriptions of how the flood water propagated during the event. While we did not have independent measurements, these narrative-based data offer valuable localized, though qualitative, information. Such information is often the only possible source of insight into local flow conditions in the absence of instrumental data and can serve to validate hydrodynamic models. Nevertheless, consistency of the reported water velocity was assessed by comparing information collected from nearby buildings.

  No changes will be made in the revised version of the manuscript.

**3. Data records**

- Figure 3: Since all the forms are connected to Form A in a 1:1 relationship, please indicate the 1:1 connection for Forms C and D in the sketch as well.

We have corrected Figure 3 according to the Reviewer suggestion.

[Figure]

**Figure 1.** Sketch of the cross section of a building depicting the measurements taken in the field ($\Delta Q$, $h_w$, $h_g$, $h_1$, $h_2$), reference level and survey point. Case with $\Delta Q$ and $h_g$ positive, and $h_1$ negative.

[Figure]

**Figure 2.** Sketch of the cross section of a building depicting the measurements taken in the field ($\Delta Q$, $h_w$, $h_g$, $h_1$), reference level and survey point. Case with $\Delta Q$ and $h_g$ negative, and $h_1$ positive.

**Variables**

- $\Delta_Q$ indicates the height difference between the elevations of the survey point and the reference level. The value is positive when the survey point is located at a higher elevation relative to the reference level point, and negative when it is below it. Figure 1 depicts the case in which this measure is positive; Figure 2 depicts the case in which this measure is negative. The measurement is taken in the field by two surveyors.

- $h_w$ indicates the external water depth outside the building, measured at the survey point.

- $h_g$ indicates the height of the first floor with respect to the survey point. This measurement is taken, for example, when the first floor is accessed via stairs. It is positive when the first floor is higher than the survey point, and negative when it is lower. Figure 1 depicts the case in which this measure is positive; Figure 2 depicts the case in which this measure is negative.

- $h_1$ represents the total height of the first floor measured relative to the $h_g$ level. It is positive when the first floor is above the $h_g$ level, and negative when it is below. Negative values clearly indicate that the first floor is a basement or semi-basement level. Figure

1 depicts the case in which $h_1$ is negative; Figure 2 depicts the case in which $h_1$ is positive.

- $\boldsymbol{h_2}$ represents the total height of the second floor measured relative to the $h_g$ level if h1 is negative, and relative to $h_1$ if $h_1$ is positive. $h_2$ is always positive, as it refers to floors above ground level.

**Survey Point and Reference Level**

- The **survey point** serves as the primary spatial reference from which vertical distances to the building floors and reference level are recorded relative to this point. Specifically, it is the exact location on the ground where the external water depth $h_w$ is measured.
- The **reference level** is a fixed elevation benchmark defined locally for each building, typically corresponding to a flat area adjacent to that building. This allows, by using a Digital Terrain Model (DTM), for all measured heights to be accurately converted into absolute elevations (e.g., the water surface elevation at the building location, provided in *FORM_A.xls*, was determined by summing $\Delta_Q$, $h_g$, and the ground elevation of the reference level).

---

## Author Comment (AC2)

**Response to Reviewer #2**

We are grateful to the Reviewer for the valuable suggestions to improve our manuscript. Below, we provide point-by-point responses to each comment.

Original review comments are in **black,** and our responses are in **blue.** *Changes introduced in the revised version of the manuscript are shown in italics.* Where necessary, references are provided using a 10-point font size.

After the flood of 2022 in the Marche region, Italy, colleagues from several Italian universities collected data on 123 damaged residential buildings, as well as on 133 affected commercial or industrial premises using standardized forms that have already been proven fit for such a purpose earlier. The paper describes the survey forms and the two datasets that are available online at: https://doi.org/10.5281/zenodo.15591850. Data can be downloaded and processed in Excel.

The joint effort of collecting damage data after such an extreme event and of providing these two unique datasets to the scientific community is noteworthy and much appreciated since it allows further development or validation of flood damage models.

The paper itself is well structured, clearly presented and well written. I have a few minor suggestions for further improvement:

- Throughout the paper, the authors use the term "business activities". I my view "premises" instead of "activities" would better describe that mostly the physical items (stocks) at the place of operation (i.e. buildings, equipment, vehicles) were surveyed, not the economic activities in terms of processes or flows.

> We thank the reviewer for the suggestion. In the revised manuscript, we will replace the term "activities" with "premises" throughout the text and dataset.

- Another term that needs some clarification, i.e. a proper definition, and some more explanation on what was collected in the field, is the term "indirect damage". For example, business interruption is not always seen as indirect damage, but as a separate category (e.g. Meyer et al, 2013: https://doi.org/10.5194/nhess-13-1351-2013).

> We acknowledge that, in the literature, the classification of business interruption varies (e.g., Meyer et al., 2013). In this study, however, we classify business interruption as an indirect damage, as in our view it represents a secondary consequence of the direct contact of flood water with the business premises. In the revised version of the manuscript (line 79), we will explicitly state the types of indirect damages collected in Form E to clarify our definition, thus avoiding possible terminological misunderstandings:
>
> > *indirect damages (i.e., delayed or secondary consequences, such as lack of usability, activity disruption, missed orders, unemployment, damage due to humidity)*

- Line 56/57: "The flood caused extensive damage to buildings and infrastructure and [...]": Do you have some official numbers on the amount of damage? Please add.

At present, no official assessment of the total damages is available.

The Marche Region has announced the availability of approximately 460 million euros as resources to support reconstruction and recovery, while journalistic sources such as La Stampa have reported damage estimates of around 2 billion euros. However, since these estimates are either partial or not official, we will not make changes to the manuscript.

References
- Marche Region, *RESOCONTO ATTIVITÀ A DUE ANNI DALL'ALLUVIONE DI SETTEMBRE 2022: https://www.regione.marche.it/portals/0/Comunicati_stampa/Slide%20Alluvione%202022_settembre_def.pdf*
- La Stampa: https://www.lastampa.it/economia/2023/12/05/news/alluvione_marche_danni_fondi_ue-13911536/

- Line 89: "which is tailored for masonry and wooden buildings" should be "... tailored to..."

We thank the Reviewer for the correction.

- Line 100: refer explicitly to Fig. 2.

We will add the reference to Figure 2 as follows:

*as illustrated in the framework diagram of Fig.2.*

- Line 112: The weekly discussions and decisions are a bit unclear to me. I thought that the three municipalities were agreed upon at the beginning of the campaign. Later in the paper (e.g. on line 120 and in line 142) it is mentioned that measures were taken to avoid duplications etc. Why was that a risk at all? And were the measures undertaken successful? Please clarify.

Line 112: Every week, a decision was made on which municipality to survey among the chosen three defined at the beginning of the campaign. Line 112 therefore refers to which of these three each team was assigned, to ensure homogeneous coverage of surveyed assets across municipalities.

Lines 120 and 142: The field campaign involved numerous researchers conducting surveys on separate occasions, which created the risk of different teams surveying the same buildings multiple times. To minimize this risk, surveyors recorded the coordinates of surveyed buildings in a shared file (see lines 133-134), which was then used in a geographic information system application on mobile devices to support the field teams by showing which assets had already been surveyed, thus helping to avoid duplication. Nevertheless, verification carried out by the external surveyor (line 142) was essential to

confirm the effectiveness of this measure and to detect duplicate entries. Based on this external review, only two housing units within the same building were found to have been entered in separate forms. These were subsequently merged into a single Form A containing two Forms B. Therefore, we can conclude that the measures undertaken were successful, although a final check remained necessary.

No changes will be made in the revised version of the manuscript.

- Line 128/129: Obviously, each building was surveyed not by an individual team member alone, but by a whole sub-team with several members having different roles. How many people were involved in one sub-team and would you recommend to keep this?

Each sub-team consisted of two or three people, as indicated in line 111. This team size was effective for the fieldwork, and we would recommend maintaining a minimum of two members, as form completion involves multiple tasks, including form filling, taking pictures, and undertaking measurements.

No changes will be made in the revised version of the manuscript.

- Line 259-261: Please add an example.

In the revised version of the manuscript, we will add an example related to Figure 5(b) at line 261:

*For instance, the high frequency of damage to systems recorded in residential buildings shown in Fig. 5(b) suggests that an effective preventive measure could be relocating electrical and heating systems to higher levels.*

- Line 285: Is this correct? Rodriguez Castro et al (2025) only report data from Belgium.

The Reviewer is right. Rodriguez Castro et al (2025) used an adapted version of the survey form for Belgium only. In the revised manuscript, we will then remove "the Netherlands" from the text.

- Line 289: What does "a significant portion" mean in terms of numbers?

According to a press release of the Marche Region published in September 2024, 3,095 damage compensation claims were submitted by private citizens and 643 by businesses. This means that the datasets cover about 4% of the damaged residential buildings and 20% of the affected business premises. At line 289 of the revised version of the manuscript, we will add these quantities and clarify that the survey only partially covered the territory affected in the Marche Region:

*According to the Marche Region, 3,095 compensation claims were submitted by private citizens and 643 by businesse (Regione Marche, 2024), indicating that the present datasets cover only 3% of the damaged residential buildings and 15% of the affected business premises. In addition, surveys were conducted in only 3 of the 14 most affected municipalities within the region.*

*Reference*
*Regione Marche, RESOCONTO ATTIVITÀ A DUE ANNI DALL'ALLUVIONE DI SETTEMBRE 2022: https://www.regione.marche.it/portals/0/Comunicati_stampa/Slide%20Alluvione%202022_settembre_def.pdf , last access: 25 August 2025, 2024.*

- Line 295-297: This is an interesting aspect. Do most of the residents only live in the affected area during the spring and summer season? If yes, please add that information earlier.

We thank the Reviewer for the observation. According to ISTAT (Istituto Nazionale di Statistica – National Institute of Statistics), 28% of the residential housing units in Senigallia are not occupied by permanent residents. This reflects the city's touristic character, with part of the population residing there only during the spring and summer seasons. In line 61, we will add the following sentence:

*Moreover, Senigallia is a coastal touristic city, where 28% of the residential dwellings are not occupied by permanent residents (ISTAT, 2021), and part of its population live there only during the spring and summer seasons.*

*Reference*
*ISTAT, Basi territoriali e variabili censuarie: https://www.istat.it/notizia/basi-territoriali-e-variabili-censuarie/, last access: 25 August 2025, 2021.*

- At the end of the paper, a brief outlook on future uses of the data and the forms would be nice. Consider shifting lines 281 to 285.

We agree with the Reviewer that providing an outlook on the future uses of the data and the forms is important. This aspect is already covered in Section 5.1 which provides an exhaustive discussion of the main potential uses of the data and forms. To avoid repetition, we have chosen to maintain the current structure of the section.

- Figure 1: Since the Figure consists of three parts, all parts should have a number or a letter (A, B, C) and a brief description in the figure caption.

We have updated Figure 1 and the caption according to the reviewer suggestion:

[Figure]

*Figure 1. (a) Marche region shown in red; (b) location of the three municipalities of Ostra, Senigallia, and Trecastelli surveyed within the Misa Basin, and the municipality of Cantiano, depicted with a green dot; (c) points representing the buildings surveyed within the three municipalities, and the flooded area of the 2022 event, as provided by the Copernicus Emergency Management Service (EMS).*

- Figure 2: Check figure caption. Do you mean "tool**s** used" instead of "tool used"?

"Tool**s**" is the correct form, and we will correct the caption of Figure 2 in the revised manuscript.

- Figure 3: The figure is clear and the structure can be found in the datasets. I was wondering whether this nested structure is feasible and ready/easy to use for data analysis. Please comment (in the discussion).

In our view, the proposed nested structure is feasible for data analysis, as it allows for a clear hierarchical organization of information and facilitates the selection of information at different level of detail and by damage type. Given this structure, users can easily navigate between aggregated summaries at the building level and records of damages to specific content types. Finally, the integration of the datasets with well-documented supporting information further strengthens its usability. The forms allow for clear interpretation of the variables present in the datasets, while the data dictionary provides clear definitions of variables and units.

In the revised version of the manuscript, at line 239, we will add the following paragraph:

*The structure of both the datasets and the forms offers a clear hierarchical organization of information, enabling analysis between multiple levels of detail, from aggregated summaries at the building level to damages occurred to specific content types. The availability of the data dictionary further facilitates the usability of the datasets by providing variables' definitions and units (see Table 1).*

- Figure 5: Please sort items in a descending/ascending order in both figures. And in 5b: The item "Damage due to high velocity" does not match with the other items since it describes a causing process, not a damaged item or activity. Please comment. The answer option "other damage" was chosen very often. Please briefly describe in the text what is included in this category. Would you recommend adding further answer options in future surveys?

We thank the reviewer for this comment. We agree that "Damage due to high velocity" refers to a damage causing process rather than a type of damage; therefore, we have removed this item from Figure 5b. The comment also helped us identify and correct an error in the original figure affecting the value of "Other damage". After correction, "Other damage" accounts for only about 10% of the damages recorded per flooded floor.

Regarding the content of the "Other damage" category, the field specifications provided by respondents are highly heterogeneous, ranging from "textbooks" to "bicycles" to damage to "wine barrels". Given this diversity, it is not possible to define a new answer category. In the revised manuscript, we will include the following paragraph at line 259:

*For residential buildings, Form B also allows respondents to select the option "other damage" when the predefined answer choices are insufficient to describe the type of damage observed. In our dataset, 10% of affected floors were reported under this category. Field specifications for these cases revealed a high degree of heterogeneity, ranging from damages to "textbooks" and "bicycles" to "wine barrels".*

In the revised Figures 5a and 5b below, the types of damage are now sorted in descending order. The caption has been revised accordingly.

[Figure]

*(a)*

[Figure]

*(b)*

[Figure]

*Figure 5. (a) Types of damage collected for business premises, disaggregated by commercial (n = 103) and industrial (n = 30) sectors, based on data from Workbooks A and C. Damage types are sorted in descending frequency order, considering commercial premises as reference. (b) Damage types collected for residential buildings floors (n = 137), based on data from Worksheet 2 of Workbook B. Damage types are sorted in descending frequency order.*

- Figure 6b: Only one dataset is shown in Fig. 6b, but two are mentioned in the figure caption. Please clarify.

> The term "datasets" in the figure caption refers to the two distinct datasets: one for residential buildings and one for economic activities. Figure 6b shows the datasets combined. We will clarify which datasets are referred to in the figure caption as follows:
>
> > *Water depth frequencies for the two datasets: residential buildings and economic activities. Water depth measurements were available for 247 out of 256 buildings surveyed.*

Thank you for this valuable effort and data.